# Roles of ERCP in the Early Diagnosis of Pancreatic Cancer

**DOI:** 10.3390/diagnostics9010030

**Published:** 2019-03-07

**Authors:** Keiji Hanada, Tomoyuki Minami, Akinori Shimizu, Motomitsu Fukuhara, Shigeki Yano, Kenji Sasaki, Masanori Koda, Kayo Sugiyama, Shuji Yonehara, Akio Yanagisawa

**Affiliations:** 1Department of Gastroenterology, Onomichi General Hospital, Onomichi 722-8508, Japan; t.minami@beach.ocn.ne.jp (T.M.); a.shimizu313@gmail.com (A.S.); motomitsu.fukuhara@gmail.com (M.F.); s.yano0319@gmail.com (S.Y.); 2Department of Pathology, Onomichi General Hospital, Onomichi 722-8508, Japan; k_sasakijaonosou@yahoo.co.jp (K.S.); m_koudajaonosou@yahoo.co.jp (M.K.); ksugi70685@gmail.com (K.S.); yonehara@eos.ocn.ne.jp (S.Y.); 3Department of Pathology, Kyoto First Red Cross Hospital, Kyoto 605-0981, Japan; yanagisa@koto.kpu-m.ac.jp

**Keywords:** pancreatic cancer, ERCP, cytology, SPACE, early diagnosis

## Abstract

It has been reported that endoscopic retrograde cholangiopancreatography (ERCP) is of value in evaluating precise pancreatograms of the pancreatic duct (PD). Recently, institutions have tended to perform magnetic resonance cholangiopancreatography (MRCP) for the diagnosis of PD due to post-ERCP pancreatitis (PEP). In small pancreatic cancer (PC), including PC in situ (PCIS) which is undetectable on cross sectional images, endoscopic ultrasonography (EUS) and MRCP serve important roles in detecting local irregular stenosis of the PD or small cystic lesions. Subsequently, ERCP and associated serial pancreatic juice aspiration cytologic examination (SPACE) obtained by endoscopic nasopancreatic drainage (ENPD) may be useful in the diagnosis of very early-stage PC. Further prospective multicenter studies are required to establish a standard method of SPACE for the early diagnosis of PC.

## 1. Introduction

It has been reported that diagnosing pancreatic cancer (PC) at an early-stage is challenging. To improve the prognosis of patients with PC, early diagnosis is essential [1,2]. In the recent Japan Pancreatic Cancer Registry analyzed by the Japan Pancreas Society (JPS), it was reported that the 5-year survival rate of cases <10 mm was 80.4%**,** and that of cases with stage 0 PC, comprising high-grade pancreatic intraepithelial neoplasia/carcinoma in situ (PCIS) was 85.8% [3]. A recent multicenter study by the Japan Study Group on the Early Detection of Pancreatic Cancer (JEDPAC) reported that the 10-year survival rate of patients with resected PC of stage 0 and I revealed a favorable prognosis [4]. However, indirect findings, such as dilatation of the pancreatic duct (PD) and pancreatic cysts, small tumor lesions difficult to visualize were observed in many cases with stage 0 and I PC [4]. It has been reported that many cases with PC arise from the branch PD [5], and PC forming a mass could easily infiltrate beyond the pancreas. Many cases with stage 0 and I PC cannot be diagnosed by imaging modalities that can detect the presence of an invasive parenchymal mass. The precise pancreatogram should be carried out for cases with indirect abnormal findings in PD. Endoscopic retrograde cholangiopancreatography (ERCP) and facilitating further examinations, such as cytology using pancreatic juice, has already been established for the early diagnosis of PC [6]. Recently, institutions have tended to perform magnetic resonance cholangiopancreatography (MRCP) for the diagnosis of PC due to the concern of post-ERCP pancreatitis (PEP). Endoscopic ultrasound-guided fine needle aspiration (EUS-FNA) has been reported to have high diagnostic accuracy for cases of early-stage PC [7]. However, it is difficult to diagnose PC without a mass-forming lesion using various image modalities. In the last 10 years, certain cases of PCIS have been diagnosed by cytodiagnosis using pancreatic juice [8,9,10]. Iiboshi et al. first reported the high accuracy in diagnosing PCIS by repeated cytodiagnosis using pancreatic juice obtained from endoscopic nasopancreatic drainage (ENPD) [11]. Recently, Satoh et al. termed this diagnostic procedure serial pancreatic-juice aspiration cytologic examination (SPACE) [12]. This review discusses the use of ERCP for the early diagnosis of PC.

## 2. Diagnostic Algorithm for Cases of Early-Stage PC in Japan

In Japan, the JPS revised the clinical guidelines (CGL) for PC in 2016. The diagnostic algorithm for PC is shown in Figure 1 [13]. Additionally, in the clinical question in this CGL, the early diagnosis of PC with a long-term prognosis is stated. First, a long-term prognosis is expected in the case of PC <1 cm. The dilatation of the pancreatic duct (PD) and cystic lesion are important indirect findings. When the small tumor lesion is difficult to detect directly by ultrasonography (US) and enhanced computed tomography (CT), endoscopic ultrasonography (EUS) or MRCP are recommended. EUS-FNA should be performed when a mass lesion is detected by EUS. When localized stenosis of the PD, caliber change, and dilatation of the branch duct are found, ERCP followed by SPACE is recommended. SPACE is considered essential for the diagnosis of PCIS [13].

## 3. Imaging Findings in Cases of Early-Stage PC

Previous studies have reported that focal branch duct dilatation, focal irregular stenosis, small cystic lesions around the stenosis of the PD, and distal dilatation of the main PD (MPD) were frequently demonstrated by EUS and MRCP [14,15]. In cases of PCIS, irregularity, non-continuous narrowing, granular defects, and dilatation were frequently observed on ERCP [14].

Recently, the JEDPAC reported 200 cases of early-stage PC (51 cases with stage 0 disease and 149 cases with stage I disease) at 14 Japanese high-volume centers [4]. The stage of PC was determined histopathologically by surgical resection. In this report, 25% of cases were symptomatic. Imaging findings, including dilatation or irregular stenosis of MPD detected by CT, MRCP, or EUS were useful for the detection of early-stage PC. Of these findings, dilatation of MPD detected by US was considered the most important initial imaging finding for the diagnosis of early-stage PC. EUS showed high visibility of stage I PC. In total, the frequency of positive image findings was lower in stage 0 than in stage I disease. Pancreatic tumors were seldom detected by imaging modalities in stage 0 cases. Preoperatively, SPACE followed by ERCP was more commonly applied than EUS-FNA (Table 1). In terms of cytological diagnosis, 72% of stage 0 cases were confirmed as malignant by SPACE, and 84% of stage I cases were confirmed as malignant by EUS-FNA (Table 2) [4].

The above observations suggest that a change in the diagnostic algorism is required, from detecting the tumorous regions by US or CT, to identify irregular stenosis or dilatation of the MPD by EUS or MRCP for the diagnosis of early-stage PC.

## 4. Pathologic Findings of PCIS

There have been recent reports on the pathologic features of PCIS. Kikuyama et al. reported cases of stage 0 disease with a high degree of fatty changes in the pancreatic parenchyma around the PCIS, which were demonstrated by enhanced CT [16]. The JEDPAC data from 200 early-stage cases also revealed that local fatty changes may be specific to early-stage PC [4] (Table 1). In certain PCIS cases, focal pancreatitis with various inflammatory cells, desmoplastic changes, fibrosis, and fatty changes were observed in the parenchyma around the PCIS. Additionally, there were PCIS cases with intraductal spread into the MPD and branch duct, and mismatch of PCIS and MPD stenosis. Focal pancreatitis, desmoplastic changes, fibrosis, and fatty change around the PCIS may be reflected as a marginal low echoic lesion by EUS [10,11,14,15].

Ikeda et al. reported the presence of PCIS in cases of invasive PC. They classified the PCIS into three types: flat (F), low papillary (LP), and mixed (FLP). Cases with the LP type showed a greater tendency than those with the F type to spread into the MPD and branch duct. Cases with the LP type may tend to change into invasive PC after spreading intraductally. By contrast, cases with the F type may tend to invade with minimal intraductal spread [17]. In our institution, 20 cases of PCIS were classified into F type and LP type according to the tumor forms. In MRCP findings, 67% of F type cases and 100% of LP type cases showed irregular MPD stenosis, and 100% of F type cases and 69% of LP type cases showed MPD dilatation. The median of MPD stenosis measured by ERCP was 4.0 mm in F type cases and 10.5 mm in LP type cases. Intraductal spread into MPD >10 mm and some skip cancer lesions in MPD were observed in LP type cases only. In addition, the LP type demonstrated long irregular MPD stenosis on MRCP and ERP (Figure 2). From these results, LP type cases may tend to spread into MPD, and its pathological features were also reflected in imaging findings, with long irregular stenosis of the MPD detected by MRCP or ERCP [18].

## 5. Cytologic Diagnosis of Cases of Early-Stage PC

It is difficult to diagnose early-stage PC cases without the presence of a formed mass as carcinoma by CT, MRI, and EUS. EUS-FNA is limited to the diagnosis of PC without mass lesions. In such cases, ERCP and its associated cytodiagnosis using pancreatic juice are expected to be useful in preoperative pathological diagnosis. However, it has been reported that the sensitivity of brushing cytology for the diagnosis of PC is not satisfactory [19].

In Japan, to improve the sensitivity of cytology, there have been case reports describing cases of early-stage PC diagnosed by SPACE. As criteria for placing an ENPD catheter, cases with localized stenosis and distal dilatation of the MPD, detected by ERCP, were predominantly selected (Table 3) [10,11,12,16,20,21,22,23,24,25,26].

Iiboshi et al. examined 20 cases (including seven cases of PCIS) with focal irregular stenosis and distal dilatation of the MPD using ENPD cytodiagnosis. A 5-French ENPD tube was placed into the MPD to collect pancreatic juice six times for 1 day. The results of SPACE for PCIS revealed a sensitivity of 100%, a specificity of 83%, and an accuracy of 95%. There was no acute pancreatitis following SPACE [11]. Other reports also demonstrated high sensitivity (100%) of SPACE for PCIS [21,26]. These observations suggest that SPACE may improve the sensitivity of cytology for PCIS.

By contrast, the ENPD method has possible associated complications, such as PEP or cholangitis. A randomized controlled trial comparing 4-French with 5-French ENPD catheters to reduce the incidence of complications suggested that an ENPD catheter with a smaller diameter reduced the incidence of PEP [27]. SPACE may have some problems, such as difficulty in identification of the lesion site, displacement or self-decannulation, processing of pancreatic juice from pancreas uncus or tail, and possibility of false positive in cytodiagnosis. In the future, the appropriate number of samples, the appropriate placing position of the ENPD tube into the MPD, and the appropriate size of the ENPD tube require evaluation in large multicenter prospective studies.

## 6. A Practical Method of SPACE for Cases of Early-Stage PC

Following the detection of abnormal findings of the MPD, such as irregular stenosis or dilatation, with a conventional ERCP, ENPD is usually performed to collect pancreatic juice samples for SPACE. Figure 3 demonstrates the ERCP catheters, guidewires, and ENPD catheters used for SPACE.

First, a 0.025-inch Radifocus guidewire is inserted to the stenosis of the MPD through an ERCP catheter. Following placement of an ERCP catheter into the stenosis of the MPD, a VisiGlide2 (Olympus Medical) or Jagwire (Boston Scientific) is inserted through an ERCP catheter. Depending on the diameter of the MPD, a 4- or 5-French ENPD tube is placed into the MPD (Figure 4). Following placement of the ENPD tube, two three-way stopcocks, through which medical staff can collect pancreatic juice safely, are additionally placed on an ENPD tube. Pancreatic juice can be collected from an ENPD tube up to six times for 1 day [28]. For the cytologic diagnosis of PCIS using pancreatic juice samples, Sasaki et al. reported that irregularly shaped nuclei were observed in cases of PCIS, with hyperchromasia, uneven chromatin distribution, and varieties of chromatin patterns among cells in the same lesion were observed in cases of PCIS only [29].

A case of PCIS in a 58-year-old female was consulted to Onomichi General Hospital due to small cystic lesions detected by US. CT revealed multiple small cystic lesions in the body of the pancreas. EUS revealed irregular MPD stenosis with round cystic lesions in the body and tail of the pancreas, and a slight low echoic lesion around the MPD stenosis (Figure 4A). MRCP revealed irregular MPD stenosis and multiple cystic lesions (Figure 4B). These observations led to the performing of ERCP and SPACE for the early diagnosis of PC. ERCP revealed irregular MPD stenosis (Figure 4C), and a 4-French ENPD tube was placed into the MPD stenosis consecutively (Figure 4D). SPACE revealed a positive result of adenocarcinoma (Figure 4E). Following distal pancreatectomy, final histologic findings revealed PCIS (Figure 4F,G). In this case, focal pancreatitis with inflammatory cells, fibrosis, and fatty change were observed in the pancreatic parenchyma around the PCIS (Figure 4F).

## 7. Evolutionary SPACE

Recently, Yokode et al. reported a clinicopathological and molecular study involving 10 cases of PCIS presenting with irregular stenosis of the MPD. The K-ras mutation was detected in nine cases, although the incidence of the overexpression of p53 and loss of *SMAD4* were low [30]. Another study of 23 cases of PCIS used targeted next-generation sequencing, which revealed that mutations of *p53, GNAS, PIK3CA, TGFBR2*, and *SMAD4* were limited [31]. These observations suggest that a dysplastic lesion in the branch PDs with abnormal expression of p53 or SMAD4 may indicate the intraductal spread of invasive PC. The value of biomarkers in pancreatic juice for the early diagnosis of PC has been addressed in various molecular targets, including microRNA [32], methylated DNA markers [33], and telomerase activity [34]. Further examination is required to establish evolutionary SPACE for the early and accurate diagnosis of PC using novel biological markers.

## 8. Summary

The present review discusses current topics of SPACE for the early diagnosis of PC. Irregular stenosis of the MPD, small cystic lesions, and a low echoic area around the stenosis of the MPD detected by EUS and MRCP, and a high degree of fatty change recognized by CT are important signs for an imaging diagnosis of early-stage PC, otherwise undetectable on cross sectional images. Further prospective multicenter studies are required to establish the standard and safety methods of SPACE for the early diagnosis of PC in the future.

## Figures and Tables

**Figure 1 diagnostics-09-00030-f001:**
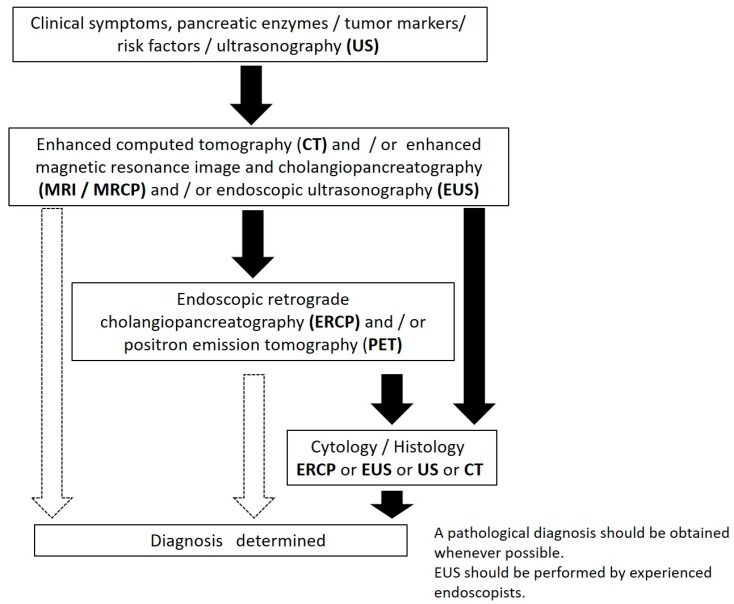
The diagnostic algorism of pancreatic cancer (PC) (from Reference [11]). The black arrow indicates frequently. The white arrow indicates sometimes.

**Figure 2 diagnostics-09-00030-f002:**
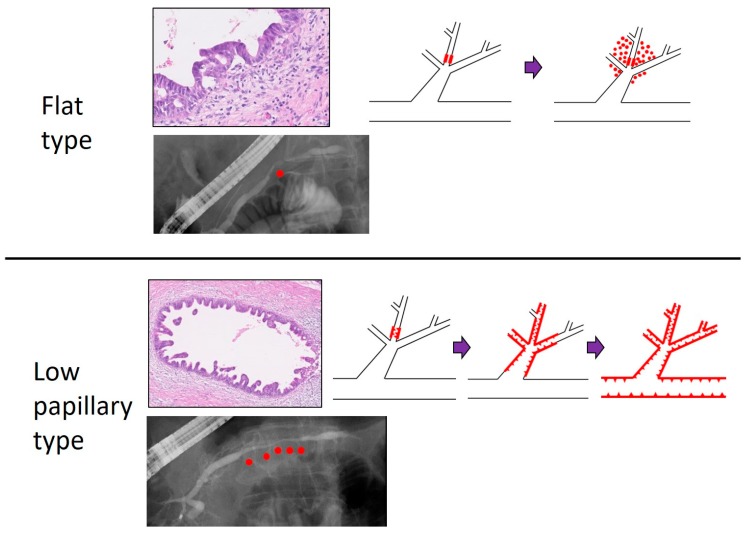
Comparison of flat and low papillary type of PC in situ (PCIS). Histologically, the case with flat type PCIS seemed to invade with little intraductal spread, whereas the case with low papillary type PCIS tended to change to invasive PC after spreading intraductally. Red dots in the ERCP image indicate the position of PCIS. Purple bar: tumor progression. Red line with triangles: tumor position. Magnification of histological slides: 10×. This study is approved by the review board of research ethics committee in Onomichi General Hospital (approval No: OJH-201862, approval date: 16 January 2019).

**Figure 3 diagnostics-09-00030-f003:**
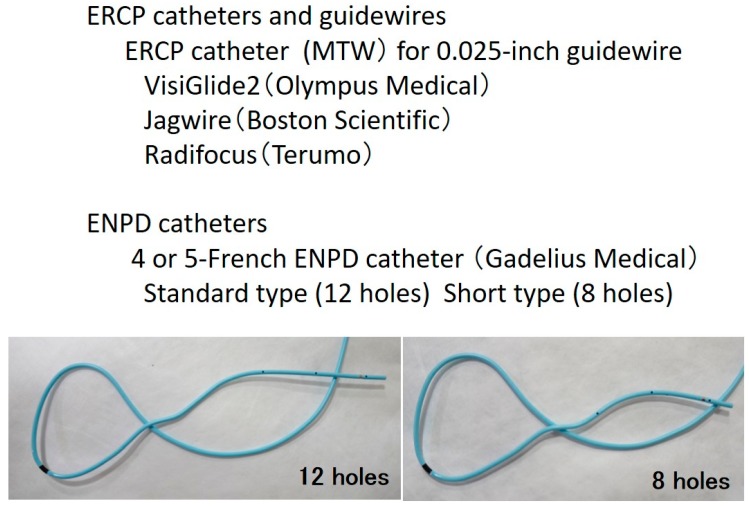
Endoscopic retrograde cholangiopancreatography (ERCP) and endoscopic nasopancreatic drainage (ENPD) catheters and guidewires for serial pancreatic juice aspiration cytologic examination (SPACE).

**Figure 4 diagnostics-09-00030-f004:**
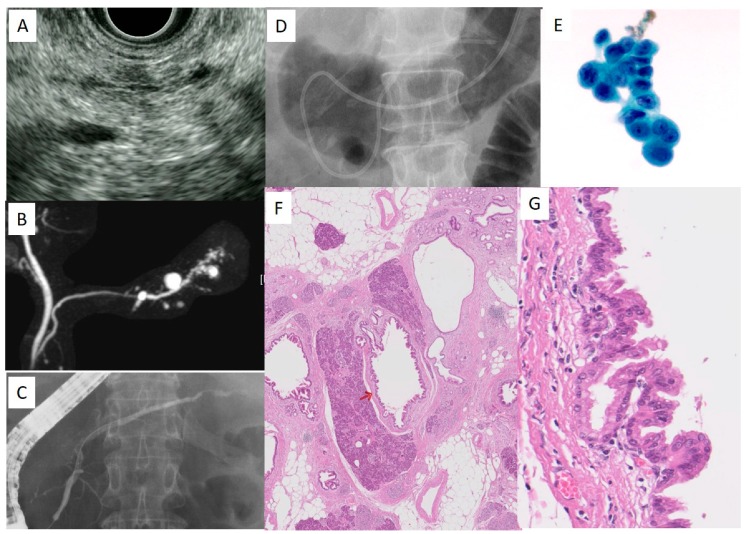
A case with PCIS (58 years old, female). (**A**): Endoscopic ultrasonography (EUS) finding in the pancreas body, (**B**): magnetic resonance cholangiopancreatography (MRCP), (**C**): ERCP, (**D**): ENPD, (**E**): Cytologic findings using pancreatic juice, (**F**): Histological findings. The red arrow indicates the location of PCIS in MPD. Magnification: 4×. (**G**): The enlarged view of PCIS in the main pancreatic duct (MPD). Magnification: 40×. This study is approved by the review board of research ethics committee in Onomichi General Hospital (approval No: OJH-201862, approval date: 16 January 2019).

**Table 1 diagnostics-09-00030-t001:** Imaging modalities and findings for the diagnosis of early-stage pancreatic cancer (PC) (from Reference 4).

Modalities	Findings	All Cases (%)	Stage 0 (%) (*n* = 51)	Stage I (%) (*n* = 149)
US		135/200 (67.5)	34/51 (66.7)	101/149 (67.8)
Findings	MPD dilatation	101/135 (74.8)	26/34 (76.5)	75/101 (74.3)
	MPD stenosis	27/135 (20)	2/34 (5.9)	25/101 (24.8)
	Tumors	71/135 (52.6)	3/34 (8.8)	68/101 (67.3)
CT		196/200 (98)	50/51 (98)	146/149 (98)
Findings	MPD dilatation	156/196 (79.6)	36/50 (72)	120/146 (82.2)
	Tumor	101/196 (51.5)	5/50 (10)	96/146 (65.8)
	Focal fatty changes	82/196 (41.8)	21/50 (42)	61/146 (41.8)
MRI		173/200 (86.5)	46/51 (90.2)	127/149 (85.2)
Findings	MPD dilatation	143/173 (82.7)	34/46 (73.9)	109/127 (85.8)
	Tumor	78/173 (45.1)	5/46 (10.9)	73/127 (57.5)
EUS		173/200 (86.5)	41/51 (80.4)	132/149 (88.6)
Findings	MPD dilatation	153/173 (88.4)	35/41 (85.4)	118/132 (89.4)
	MPD stenosis	98/173 (56.6)	28/41 (68.3)	70/132 (53)
	Tumor	132/173 (76.3)	10/41 (24.4)	122/132 (92.4)
ERCP		141/200 (70.5)	47/51 (92.2)	94/149 (63.1)
Findings	MPD dilatation	114/141 (80.9)	39/47 (83)	75/94 (79.8)
	MPD stenosis	112/141 (79.4)	39/47 (83)	73/94 (77.7)
FDG-PET		61/200 (30.5)	11/51 (21.6)	50/149 (33.6)
	FDG accumulation	31/61 (50.8)	1/11 (9.1)	30/50 (60)

MRI: Magnetic Resonance Imaging. FDG-PET: Fluorodeoxyglucose-positron emission tomography.

**Table 2 diagnostics-09-00030-t002:** Cytological diagnosis of early-stage PC (from Reference 4).

Cytology		All Cases (*n* = 200)	Stage 0 (*n* = 51)	Stage I (*n* = 149)
Cytology by ERCP		79/141 (56)	36/47 (77)	48/94 (51)
Confirmation of Malignancy	Brush	30/62 (48)	6/14 (43)	24/43 (56)
	ENPD	55/79 (70)	26/36 (72)	29/48 (60)
Cytology by EUS-FNA		69/200 (35)	6/51 (12)	63/149 (42)
Confirmation of Malignancy		54/69 (78)	1/6 (17)	53/63 (84)

**Table 3 diagnostics-09-00030-t003:** Lists of early-stage PC case diagnosed by serial pancreatic juice aspiration cytologic examination (SPACE).

Authors/Years	Number of Cases	Depiction of Tumor	MPD Findings	Methods of Collection of Pancreatic Juice	Number of Collection of Pancreatic Juice	Pathological Stage
Kimura et al. 2011 [26]	3	No	Stenosis and distal dilatation	ENPD	3	PCIS
Iiboshi et al. 2012 [11]	7	No	Stenosis and dilatation	ENPD	6	PCIS
Mikata et al. 2013 [21]	2	No	Stenosis and distal dilatation	ENPD	6	PCIS
Shindo et al. 2014 [20]	1	No	Stenosis and distal dilatation	ERCP brushing and ENPD	4	PCIS
Maehira et al. 2014 [22]	1	No	Stenosis and distal dilatation	ERCP brushing and ENPD	2	PCIS
Kikuyama et al. 2015 [16]	2	No	Stenosis and dilatation	ENPD	6	PCIS
Ohtsubo et al. 2017 [23]	1	No	Stenosis and distal dilatation	ENPD	6	PCIS and minimally invasion
Kaneko et al. 2017 [24]	1	No	Stenosis and distal dilatation	ENPD	6	PCIS
Minami et al. 2017 [25]	11	No	Stenosis and distal dilatation	ENPD	6	PCIS
Miyata et al. 2017 [10]	1	No	Stenosis and distal dilatation	ENPD	6	PCIS
Sato et al. 2017 [12]	1	No	Stenosis and distal dilatation	ENPD	6	PCIS

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
