# Peer review of "Roles of ERCP in the Early Diagnosis of Pancreatic Cancer"

_diagnostics, 2019, doi:10.3390/diagnostics9010030_

Round 1

Reviewer 1 Report

The changes made since the previous iteration provide a more structured and easier to read review. 

Author Response

Comment 1

The changes made since the previous iteration provide a more structured and easier to read review. 

Response 1

Thank you very much for your kind comment.

Reviewer 2 Report

In this manuscript, the authors have discussed how ERCP and SPACE can be used to detect early stage pancreatic cancer. The techniques are well described and give an idea of why they may be advantageous to use. The authors should also state some disadvantages/limitations of these techniques. In addition, I have some comments below:

1.      The authors should create a list of abbreviations that they are using. This will be useful to the reader.

2.      In the abstract PC means Pancreatic cancer I assume? The authors should state pancreatic cancer (PC); this will ensure readers know what the abbreviation stands for.

3.      The introduction should include why it is difficult to diagnose PC at early stage.

4.      The authors should add why ERCP is accepted as good way to diagnose PC early in the introduction.

5.      EUS-FNA should be expanded in line 50. It should state EUS-Fine needle aspiration

6.      In line 62, authors mention the number of stage 0 and stage 1 cases reported in JEDPAC. The authors should also mention what method was used to diagnose these cases. Also will be good to elaborate how they characterize stage 0 and stage 1 PC cases.

7.      Are there any limitations to using SPACE or ERCP? Is there any case where false positives are reported?

Author Response

In this manuscript, the authors have discussed how ERCP and SPACE can be used to detect early stage pancreatic cancer. The techniques are well described and give an idea of why they may be advantageous to use. The authors should also state some disadvantages/limitations of these techniques. In addition, I have some comments below:

Comment 2

The authors should create a list of abbreviations that they are using. This will be useful to the reader.

Response 2

This time, we created a list of abbreviations in this manuscript.

Comment 3

In the abstract PC means Pancreatic cancer I assume? The authors should state pancreatic cancer (PC); this will ensure readers know what the abbreviation stands for.

Response 3

We have already described the first of spelling “pancreatic cancer” to abbreviation “PC” in line 4 of the abstract.  

Comment 4

The introduction should include why it is difficult to diagnose PC at early stage.

Response 4

We included the description about the difficulty to diagnose PC at an early stage in the introduction.

Comment 5

The authors should add why ERCP is accepted as good way to diagnose PC early in the introduction.

Response 5

We included the description the value of ERCP to diagnose PC at an early stage in the introduction.

Comment 6

EUS-FNA should be expanded in line 50. It should state EUS-Fine needle aspiration

Response 6

We have already described the first of spelling “Endoscopic ultrasound-guided fine needle aspiration” to abbreviation “EUS-FNA” in the introduction.

Comment 7

In line 62, authors mention the number of stage 0 and stage 1 cases reported in JEDPAC. The authors should also mention what method was used to diagnose these cases. Also will be good to elaborate how they characterize stage 0 and stage 1 PC cases.

Response 7

We added the sentence as follows. “The stage of PC was determined histopathologically by surgical resection.”

Comments 8

Are there any limitations to using SPACE or ERCP? Is there any case where false positives are reported?

Response 8

We described some problems about SPACE that should be dissolved in chapter 5.

Reviewer 3 Report

Hanada et al have summarized the diagnostic algorithm used in pancreatic cancer detection, especially focusing on early detection and detection before formation of mass lesions. The review is well organized and lucid. Minor English editing is suggested before it is accepted.

Author Response

Comment 9

Hanada et al have summarized the diagnostic algorithm used in pancreatic cancer detection, especially focusing on early detection and detection before formation of mass lesions. The review is well organized and lucid. Minor English editing is suggested before it is accepted.

Response 9

This time, the manuscript has been carefully reviewed by an experienced editor whose first language is English and who specializes in editing papers written by scientists whose native language is not English.

Round 2

Reviewer 2 Report

The authors have addressed my concerns and I support publication of this manuscript

This manuscript is a resubmission of an earlier submission. The following is a list of the peer review reports and author responses from that submission.

Round 1

Reviewer 1 Report

The authors present a review on the current state of early detection in pancreas cancer and focus on SPACE. Although, this topic is very important and the content of the manuscript is very relevant to the field, the review is not well-written and is missing elements of structure. Some of the issues that should be addressed include: 

The title of the manuscript does not reflect the actual content as it is more specific than what is discussed in the body of the manuscript. 

Section 5 should have a table to summarize the various studies mentioned. 

Restructuring this review, and allowing the content to be more focused and concise would make for a more compelling paper. 

Reviewer 2 Report

This is a review suggesting SPACE as a diagnostic method that can detect pancreatic cancer at pre-malignant stage (stage 0). The review examines all the evidence around diagnostic imaging and cytology methods for early diagnosis of pancreatic adenocarcinoma. Although the evidence is very limited, the authors suggest the development of a new diagnostic tests to identify pancreatic cancer early.

The way that review is written is hard to follow and understand the benefit of the combined imaging methods the authors suggest. It would be helpful if there was a clear flowchart showing the current practice and how SPACE will improve diagnosis.

Also, what the review misses is a section outlining the limitations of these clinical tests. For example, can they be easily introduced in endoscopy units? What training does the workforce need? Also, what symptoms would prompt a doctor to recommend such a diagnostic test? It is important that the authors have a section discussing what populations are likely to benefit.